# The Role of LncRNAs in Radio- and Chemoresistance of Glioblastoma: Prognostic or Therapeutic?

**DOI:** 10.3390/curroncol32100539

**Published:** 2025-09-27

**Authors:** Elisa Tremante, Ana Belén Díaz Méndez, Maria Giulia Rizzo

**Affiliations:** Department of Research, Advanced Diagnostics and Technological Innovation, Translational Oncology Research Unit, IRCCS Regina Elena National Cancer Institute, 00144 Rome, Italy; anabelen.diaz@ifo.it (A.B.D.M.); maria.rizzo@ifo.it (M.G.R.)

**Keywords:** lncRNA, glioblastoma, radio-resistance, chemo-resistance, prognostic, therapeutic

## Abstract

This review highlights the role of lncRNAs in tumor progression and therapy resistance, focusing particularly on their ability to drive the transition from reversible epigenetic adaptations to stable genetic changes in resistant cancer cells. lncRNAs not only modulate gene expression and signaling pathways through interacting with miRNAs, DNA, RNA, and proteins, but also actively contribute to the reprogramming of the cellular environment. This action facilitates the shift from early adaptive, often transient, epigenetic mechanisms of resistance to more permanent genetic alterations that support tumor evolution and sustained therapy resistance. Their widespread expression and deregulation in tumors suggest their strong potential as prognostic biomarkers and as therapeutic targets, especially in combination with traditional or next-generation treatments.

## 1. Introduction

One of the major challenges in cancer treatment is the ability of cancer cells to acquire resistance to therapies. Traditionally attributed to genetic mutations that enable cancer cells to evade therapeutic pressure, emerging evidence underscores the critical contribution of non-genetic mechanisms, particularly those involving transcriptional reprogramming [1]. Key signaling pathways such as PTEN/pAKT and WNT/β-catenin have been recurrently implicated in therapy resistance across multiple cancer types [2]. Understanding the underlying mechanisms of treatment resistance, including specific pathways and molecular players, is essential. While both genetic and non-genetic mechanisms offer valuable insights, further investigation is especially needed into cellular tolerance and persistence. The principal epigenetic mechanisms implicated in cancer regulation include DNA methylation, histone modifications, and ncRNA, which play pivotal roles in modulating gene expression without altering the underlying DNA sequence [3]. DNA methylation involves the covalent addition of a methyl group to the 5-position of cytosine residues, predominantly within CpG dinucleotides. This modification induces conformational changes in DNA, influences genomic stability, reshapes chromatin organization, and affects DNA-protein interactions [4]. Histone modifications, including acetylation, methylation, phosphorylation, ubiquitination, and succinylation, are essential for maintaining chromatin integrity and for regulating processes such as transcription, replication, DNA repair, and recombination [5]. The ncRNA contributes to cancer epigenetics by guiding chromatin-modifying complexes to specific genomic loci, thereby altering the chromatin structure and gene expression. Among the diverse groups of non-coding RNAs (ncRNAs) that regulate epigenetic landscapes in cancer, long non-coding RNAs (lncRNAs) have recently emerged as key players. Alongside microRNAs (miRNAs) and small interfering RNAs (siRNAs), lncRNAs interact with DNA, RNA, and proteins to form a complex and dynamic regulatory network. Increasing evidence shows that ncRNAs modulate multiple layers of epigenetic regulation—including DNA methylation and histone modifications—positioning them as critical coordinators of the cancer epigenetic landscape [6]. Recent studies have identified long non-coding RNAs (lncRNAs), protein non-coding transcripts longer than 200 nucleotides, as crucial regulators of therapy resistance and potential prognostic biomarkers across diverse tumor types [7]. In malignant brain tumors, lncRNAs modulate resistance to radiotherapy, chemotherapy, and targeted agents, in part by influencing epithelial–mesenchymal transition (EMT), cytoskeletal remodeling, and extracellular matrix interactions—which are core components of the tumor microenvironment [8,9]. In terms of function, lncRNAs interact via the epigenetic regulation of DNA methylation, histone modification, chromatin remodeling, or the binding to other molecules that act as competing endogenous RNAs (ceRNAs), by either promoting or suppressing tumor progression depending on the context [10]. When acting as oncogenes or tumor suppressors, lncRNAs influence tumor progression by shaping the cancer epigenome and inducing epigenetic modifications that affect proliferation, apoptosis, EMT, migration, invasion, metastasis, cancer stemness, and chemoresistance [11]. LncRNAs exert their gene regulatory functions through a variety of mechanisms, including chromatin modification and binding to proteins such as transcription factors. Among them, their ability to sequester microRNAs (miRNAs) is particularly relevant in tumor progression and both primary and acquired resistance to treatments [12]. Many lncRNAs contain multiple miRNA response elements (MREs), enabling them to fine-tune miRNA availability and, thus, influence key cellular processes such as development, differentiation, and disease progression. In cancers like glioblastoma (GBM) and therapy-resistant brain tumors, lncRNAs are often dysregulated and can promote tumor growth by sponging tumor-suppressive miRNAs or, less commonly, act as tumor suppressors. Non-genetic mechanisms of adaptability enable tumor cells to tolerate and survive therapeutic pressure [13]. These adaptive responses, modulated by lncRNAs, may facilitate the transition to stabilize resistance during which genetic mutations may arise and consolidate a permanently resistant phenotype. Understanding these dynamic processes opens new avenues for reprogramming tumor cells, thereby enhancing their susceptibility to treatment. The mechanisms underlying genetic alterations and epigenetic adaptations, including the stabilization of epigenetic modifications, are critical processes that contribute to therapeutic resistance. Among the various malignancies requiring immediate intervention, glioblastoma is particularly striking due to its highly aggressive and localized phenotype. Therefore, it is vital that the molecular pathways involved in resistance mechanisms are thoroughly characterized and rapidly targeted to enhance treatment efficacy. Emerging challenges include understanding the roles of lncRNAs and their dysregulated pathways in cancer [14,15,16,17]. GBM is the most common malignant brain tumor, which accounts for 14.6% of all tumors [18]. The prognosis for patients with GBM remains extremely poor, with a 5-year survival rate ranging from 2% to 10% [19]. This aggressive cancer poses significant challenges for both diagnosis and treatment, largely due to its anatomical location, diffuse and infiltrative growth pattern, its impact on brain function, and its complex biological features [20]. Standard clinical management typically involves surgical resection followed by a combination of chemotherapy and regional fractionated radiotherapy [21]. Temozolomide (TMZ), an alkylating agent, is the most commonly used and approved chemotherapeutic for both first- and second-line treatment in GBM. However, most patients exhibit either intrinsic or acquired resistance to therapy, and almost all eventually experience disease progression or recurrence. LncRNAs exert molecular effects and interactions involved in radio- and chemoresistance across various malignant CNS tumor types. Recent studies have highlighted the dysregulation of non-coding RNAs (ncRNAs) in many human cancers, including glioblastoma, medulloblastoma, meningioma, and pituitary adenoma [22,23,24,25]. These studies often emphasize the potential prognostic and therapeutic roles of specific lncRNAs in these tumors, particularly glioblastoma. Some studies have also explored the interplay between microRNAs and lncRNAs, offering further insight into the pathways involved in tumor progression and treatment resistance. Notably, the most extensively studied lncRNAs in malignant brain tumors have often been previously investigated in the context of other cancers. This review aims to discuss and summarize the current literature from the last five years regarding lncRNAs and malignant brain tumors, which particularly focus on their therapeutic and prognostic relevance in the context of therapy resistance. We highlight lncRNAs as a valuable source of information due to their roles in radio- and chemoresistance, with the potential to be integrated into clinical practice in order to improve the management of patients with malignant brain tumors and support personalized medicine.

## 2. LncRNAs

In the Appendix A “lncRNAs, miRNA sponge, pathway involved” lists the lncRNAs with oncogenic or tumor suppressor roles. In the following sections, they will be described and categorized based primarily on their prognostic or therapeutic value.

### 2.1. Prognostic Role of LncRNAs

The lncRNAs play a role in transcription and genetic modification. Moreover, its function is associated with tumor dysregulation. In malignant brain tumors, particularly in glioblastomas, lncRNAs dysregulation seems to be linked to prognosis, especially in relation to patient survival and resistance prediction (Table 1).

### 2.2. MAPK/ERK Pathway

LINC00461 (ECONEXIN) has been identified as a prognostic oncogene involved in tumor progression, also in gliomas, affecting cell proliferation, migration, and invasion via MAPK/ERK, PI3K/AKT, and other signaling pathways [26]. In 2020, Ting Tang studied the transmembrane phosphatase with tensin homology pseudogene 1 (*TPTEP1*) in gliomas. They postulated that its downregulation in high-grade gliomas is associated with better patient prognosis. Experiments in vitro and in vivo confirmed their hypothesis of its involvement in cell stemness and radioresistance suppression in gliomas. TPTEP1 promotes *MAPK14* expression by antagonizing miR-106a-5p binding and activating the P38 MAPK signaling pathway [27]. SNHG12 acts as a sponge for miR-129-5p, leading to upregulation of *MAPK1* and *E2F7*, which confers TMZ resistance in GBM cells. SNHG12 overexpression correlates with poor survival in GBM patients treated with TMZ [28].

### 2.3. PI3K/AKT Pathway

Knockdown of MSC-AS1 suppressed cell growth and chemoresistance of glioma cells to TMZ by regulating the miR-373-3p/CPEB4 axis in vitro and in vivo through activation of the PI3K/Akt pathway. Li C and colleagues have evidence supporting the potential role of MSC-AS1 as a prognostic biomarker in glioma patients undergoing TMZ-based chemotherapy [29]. XLOC013218 (XLOC) is upregulated in TMZ-resistant cells linked to poor prognosis of gliomas. Overexpression of XLOC increases TMZ resistance, promotes proliferation, and inhibits apoptosis. *PIK3R2* is identified as its target, with XLOC enhancing Sp1 binding to the *PIK3R2* promoter, boosting its expression. In turn, this activates the PI3K/AKT pathway, driving TMZ resistance and cell growth. The XLOC/Sp1/PIK3R2/PI3K/AKT axis plays a key role in GBM TMZ resistance [40].

### 2.4. Wnt/β-Catenin Signaling Pathway

Assuming that *MIR155HG* is overexpressed in TMZ-resistant glioblastoma cells, the study by He X and Sheng J demonstrated that knocking it down enhances treatment sensitivity by inhibiting the Wnt/β-catenin pathway [31]. By highlighting the overexpression of *SOX2OT* in cells resistant to TMZ treatment, Liu B et al. investigated the underlying mechanism of action. Their data demonstrate that lncSOX2OT exerts an oncogenic role through the Wnt5a/β-catenin signaling pathway, thus serving as a prognostic marker as well as having the potential as a therapeutic lncRNA [32,50].

### 2.5. Cell Cycle Regulation

E2F1 could target the promoter region of LINC00021, which represses p21, thus promoting glioblastomagenesis [45]. HOTAIR Hox transcript antisense intergenic RNA (HOTAIR) is located in the HOXC locus and functions as a transcriptional repressor. Elevated HOTAIR expression has been implicated in the progression of multiple primary tumors, promoting invasion and metastasis [46,51]. HOTAIR regulatory elements function as silencers to modulate glioma cell sensitivity to TMZ through long-range regulation of the targets, *CALCOCO1 and ZC3H10* [52]. HOTAIR is identified as a negative prognostic factor in breast and colon cancers, as well as in the liver, colon, and laryngeal squamous cell carcinomas [53,54,55]. Its oncogenic role entails regulating various target genes through epigenetic mechanisms as well as sponging activity, which influences critical cellular and signaling pathways associated with metastasis and drug resistance. HOTAIR promotes malignant progression and poor prognosis in glioma patients, with its pro-oncogenic activity linked to cell cycle modulation [30,56].

### 2.6. miRNA Sponging and ceRNA Activity

Deguchi et al. demonstrated that ECONEXIN, located predominantly in the cytoplasm, regulates *TOP2A* by sponging miR-411-5p via two binding sites, thus promoting tumor progression [34]. NCK1-AS1 can increase drug resistance in glioma cells to TMZ by modulating the miR-137/TRIM24 pathway [35]. According to Tian W and colleagues, LINC01123 serves as a sponge for miR-151a and upregulates CENPB expression, increasing the radioresistance of glioma cells in vitro and in vivo [36]. HOXA-AS2 knockdown resulted in a corresponding drop in IGF1 expression, consistent with indirect regulation mediated by miR-302a-3p [48]. MAFG-AS1 reduces the radiosensitivity of GBM cells via the miR-642a-5p/Notch1 axis [41]. While HOTAIR expression is typically low in normal brain tissue, it is significantly upregulated in gliomas, particularly in temozolomide-resistant GBMs. In a bioinformatics analysis performed by Lan T and colleagues, it was observed that HOTAIR expression correlates with drug resistance in TMZ-resistant GBM cell lines, showing data where HOTAIR (acting as a ceRNA) regulates β-catenin and MGMT by sponging miR-214, thus increasing TMZ sensitivity in resistant GBM cells [33]. Previous studies have highlighted the potential prognostic value of HOTAIR [57] or have either focused on other mechanisms, such as extracellular vesicle (EV)-mediated transfer of long non-coding RNAs. Studies suggest that serum-derived extracellular vesicles (EVs) transport HOTAIR to GBM cells, where it enhances tumor progression and resistance to TMZ by binding miR-526b-3p and increasing the EVA1 expression [37,38].

### 2.7. Tumor Progression

In TMZ-resistant GBM cells, HOTAIR is highly expressed and by reducing its levels inhibits cell growth, migration, invasion, and epithelial-to-mesenchymal transition (EMT) [58]. Zhang J showed that the interplay between the HOTAIR/miR-125 and miR-125/HK2 pathways could potentially reveal new targets for the prevention and treatment of TMZ-resistant GBM [39]. MEG3 (maternally expressed gene 3) is an imprinted gene located on chromosome 14, which also contains long non-coding RNAs, miRNAs, and nucleolar RNAs [47]. When overexpressed, MEG3 inhibits EMT in cervical cancer cells, whereas in hepatocellular carcinoma, it enhances the migration and invasion of cancer cells by activating the transcriptional activity of the tumor suppressor p53. Additionally, MEG3 plays a role in regulating cell proliferation, apoptosis, and the EMT process in pituitary tumors, positioning it as a promising target for the treatment of pituitary tumors [49]. According to several authors, LINC00520 contributes to TMZ chemoresistance and acts as a prognostic marker [59]. FOXD3-AS1 is critically involved in GBM cell survival and resistance to TMZ, highlighting its potential as a prognostic biomarker for treatment response and a possible target for GBM therapy [42].

### 2.8. Metabolic Reprogramming and Stress Response

ZBED3-AS1 downregulation and THBD activation are linked to an increased TMZ resistance and glycolysis in GBM cells [43]. ATXN8OS mediates ferroptosis and regulates TMZ resistance in gliomas via the ADAR/GLS2 pathway, thereby acting as a tumor suppressor. ATXN8OS in GBM is associated with resistance to TMZ, and thus, its low expression serves as a prognostic biomarker for TMZ-based chemotherapy [44].

### 2.9. LncRNAs as Therapeutic Targets

Based on these properties, lncRNAs represent promising therapeutic targets, as their modulation could restore normal gene regulation, alter tumor-promoting pathways, and potentially overcome resistance to conventional treatments (Table 2).

### 2.10. NF-κB Pathway

LIFR-AS1 is a tumor suppressor and promotes apoptosis by modulating the miR-4262/NF-κB pathway in gliomas [60]. Tang G and colleagues showed that LINC01057 is overexpressed in GBM, especially in the mesenchymal subtype (MES), and promotes MES differentiation via activating NF-κB signaling [79]. LINC01057 regulates NF-κB signaling to promote mesenchymal differentiation in GBM, suggesting it could be a potential target for therapeutic intervention in the MES subtype of GBM [79]. Another paper suggests that the role of H19 could depend on its ability to recruit and activate NF-κB signaling, which may represent a novel therapeutic target for TMZ-resistant gliomas [61]. Recent evidence supports the role of lncRNA H19 as a significant biomarker for the diagnosis and therapeutic intervention of gliomas. This may be attributable to its upregulated expression in glioma tissues and its involvement in the regulation of cellular proliferation and metastatic processes [80]. H19 has been implicated in the development of resistance to both chemotherapy and radiotherapy, positioning it as a potential target for therapeutic modulation. Elevated H19 expression correlates with treatment-resistant phenotypes, promoting resistance through the induction of anti-apoptotic genes such as *BCL-2*, multidrug resistance-associated proteins including MDR1/2, and the enhancement of autophagic mechanisms. The functional inhibition of H19 restores chemosensitivity and radiosensitivity in resistant tumor cells, while exosome-mediated transfer of H19 contributes to the horizontal propagation of resistant traits within the tumor microenvironment. These findings support the integration of H19-targeted approaches alongside conventional therapies to enhance anti-tumor efficacy and improve clinical outcomes [80,81,82].

### 2.11. PI3K/Akt/mTOR Pathway

The KCNQ1OT1/miR-761/PIM1 axis plays a critical role in modulating chemoresistance in gliomas and represents a promising therapeutic target for overcoming TMZ resistance. KCNQ1OT1 promotes resistance by sponging miR-761, thereby enhancing PIM1 expression and subsequently upregulating downstream effectors such as MDR1, c-Myc, and Survivin [62]. The lncRNA CRNDE influences cell proliferation, apoptosis, migration, invasion, and chemoresistance in medulloblastoma by binding to and suppressing miR-29c-3p, indicating its potential utility in therapeutic strategies [83]. Furthermore, CRNDE modulates autophagy via the PI3K/Akt/mTOR pathway and regulates ABCG2 expression, supporting its candidacy as a novel target for overcoming TMZ resistance in GBM [68]. LINC01410 enhances TMZ sensitivity by inhibiting the PTEN/AKT pathway via miR-370-3p [69]. A study suggests that two lncRNAs, ARFRP1 and RUSC2, regulate five genes (*IRS1*, *FOXG1*, *GNG2*, *RUNX2*, and *CACNA1E*) involved in key signaling pathways such as AMPK, AKT, mTOR, and TGF-β, which influence autophagy and contribute to TMZ resistance. This newly identified lncRNA-associated ceRNA network in glioblastoma offers new insights into TMZ resistance and potential therapeutic targets [84]. MAGI2-AS3 suppresses tumor growth and enhances TMZ’s effectiveness against gliomas by inhibiting the Akt pathway, thereby reversing TMZ resistance [85]. 

### 2.12. Autophagy

Linc-RA1 inhibits autophagy activation, affecting glioma cell radioresistance in addition to suggesting its use as a biomarker and therapeutic target [70]. CRNDE modulates autophagy via the PI3K/Akt/mTOR pathway and regulates *ABCG2*, affecting TMZ resistance in GBM [68]. CASC2 expression is reduced in gliomas, which leads to the upregulation of miR-193a-5p and subsequent downregulation of mTOR. This cascade promotes protective autophagy, contributing to TMZ resistance. Consequently, autophagy inhibition has been shown to enhance TMZ therapeutic efficacy [86].

### 2.13. Wnt/β-Catenin Pathway

RMRP promotes TMZ resistance by activating the Wnt/β-catenin pathway [71]. LINC00511, which is localized in the cytoplasm of GBM cells, modulates Wnt/β-catenin signaling by acting as a molecular sponge for miR-126-5p. The Wnt/β-catenin pathway genes, including *DVL3*, *WISP1*, and *WISP2*, were found to be targets of miR-126-5p. The expression of miR-126-5p reduced the resistance of GBM cells to TMZ, thus confirming the potential use of this lncRNA as a therapeutic target [72].

### 2.14. miRNA Sponge Networks (ceRNA Axes)

BC200 promotes glioblastoma oncogenicity and TMZ resistance through miR-218-5p modulation [65]. PSMB8-AS1 acts as a miR-22-3p sponge to mediate *DDIT4* expression and regulate glioblastoma progression [63]. The lncRNA TMEM161B-AS1 sponges miR-27a-3p, impacting the FANCD2/CD44 signaling axis. The effects are visible in GBM resistance, suggesting promising therapeutic targets for glioma treatment. This lncRNA could also be useful in combination with immunotherapy [87]. LINC00883 binds to miR-136, preventing its downregulation of NEK1. In U251 cells with high drug resistance, LINC00883 overexpression increased MRP expression, reduced apoptosis, and promoted proliferation. The depletion of LINC00883 enhanced tumor-suppressive and anti-chemoresistance effects by increasing miR-136 and inhibiting NEK1 [88]. TP73-AS1 lower expression is linked to TMZ resistance in U87MG cells, with its knockdown reducing TMZ sensitivity, indicating TP73-AS1’s role in regulating drug response [89]. LncRNA HCG11 regulates glioma cell proliferation, apoptosis, and drug resistance through the miR-144/COX-2 axis, and its down-regulation is involved in the resistance to Imatinib, VP-16, and TMZ [90]. LINC00957 regulates the miR-17-5p/NPNT axis; its effects on cell cycle and migration in GBM are confirmed by the authors [91]. 

### 2.15. TGF-β/Smad Pathway

In glioma cells, TGF-β induces the expression of lncRNA-MUF, which is associated with poor prognosis in GBM patients. Reducing lncRNA-MUF levels limits tumor cell growth, migration, and invasion, and increases sensitivity to TMZ-induced cell death. It also disrupts the TGF-β-mediated activation of *Smad2/3*. While acting as a molecular sponge for miR-34a, lncRNA-MUF enhances Snail1 expression. These findings suggest that targeting lncRNA-MUF could offer a new therapeutic approach for GBM [78]. RP11-838N2.4 enhances sensitivity to TMZ by down-regulating miR-10a, leading to an increase in the EphA8 expression of both in vitro and in vivo models, while suppressing TGF-β signaling activity at the same time. From a clinical perspective, reduced expression in GBM patients correlates with higher tumor recurrence and poorer outcomes, indicating significant potential as both a prognostic marker and therapeutic target [92].

### 2.16. CHK1 and DNA Repair Pathway

The CCAT2/miR-424/Chk1 signaling axis represents a potential therapeutic target for enhancing the efficacy of chemotherapy treatment in patients with glioma [93]. In medulloblastoma, multiple studies identify RBM5-AS1 as a promoter of stem-like properties and the cause of resistance to radiation. In vivo studies demonstrated that reducing RBM5-AS1 levels suppresses tumor growth and enhances radiosensitivity, while its overexpression lessens radiation-triggered apoptosis and DNA damage in medulloblastoma cells through sirtuin 6 (SIRT6). Therefore, targeting RBM5-AS1 could be a promising approach to counteract radiotherapy resistance in medulloblastoma [73]. The CHK1 inhibitor SRA737 abolishes the TMZ-induced structural remodeling of LINC01956 and subsequent MGMT upregulation, consequently resensitizing TMZ-resistant MGMT promoter-methylated GBM to TMZ [75]. A study published in *Cancer Letters* illustrated that EPIC-0628, a small-molecule inhibitor that selectively disrupts the HOTAIR-EZH2 interaction, promotes ATF3 expression. This agent silences MGMT expression and enhances TMZ efficacy in GBM, consequently inducing cell cycle arrest by upregulating CDKN1A and impairing DNA repair via the ATF3-p38-E2F1 pathway, suggesting the potential of the combination treatment with TMZ [76].

### 2.17. IGF Signaling Pathway

Similarly, NCK1-AS1 acts as an oncogenic lncRNA that contributes to glioma cell proliferation, radioresistance, and chemoresistance through the miR-22-3p/IGF1R ceRNA network. Wang and colleagues suggest using NCK1-AS1 to improve glioma treatment (radiotherapy and chemotherapy) [74]. LINC01123 promotes glioma cell radioresistance by sponging miR-151a and increasing CENPB levels [76]. Inhibiting *OIP5-AS1* enhances miR-129-5p expression, which reduces TMZ resistance in GBM cells by targeting *IGF2BP2* [94].

### 2.18. Cell Cycle Regulation

LINC00174 can sponge miR-138-5p and down-regulate its expression. This, in turn, leads to the upregulation of the target sex-determining region Y (SRY)-box 9 protein (SOX9), which could influence glioma proliferation and act in the downregulation of apoptosis [95]. DARS1-AS1 interacts with YBX1 to stabilize mRNAs promoting expression of *E2F1* and *CCND1*, which control the G1-S transition, and stabilizes FOXM1, vital for glioma stem cell self-renewal and DNA repair, hence contributing to GBM radioresistance [77]. MIR210HG is upregulated under hypoxia and interacts with *OCT1* to modulate hypoxic GBM, leading to poor prognosis [96]. Targeting SPI1 blocks the transcription of MIR222HG, which promotes the proneural-to-mesenchymal transition of glioma stem cells and immunosuppressive macrophage polarization, thereby reducing radioresistance and the immunosuppressive microenvironment in GBM [97]. TP53TG1 impacts glioma proliferation, colony formation, autophagy, and radioresistance through the miR-524-5p/RAB5A axis [64]. LINC00942 promotes *SOX9* expression by interacting with *TPI1* and *PKM2*, driving self-renewal and TMZ resistance in GBM cells [98].

### 2.19. Other Mechanisms 

EPIC1 exhibits its function by targeting Cdc20 in glioma cells [63]. Exosome-transferred lncRNA TALC alters the glioblastoma microenvironment, reducing sensitivity to TMZ and suggesting combination therapies to overcome resistance [99]. PDIA3P1 is upregulated in TMZ-resistant GBM cell lines; targeting p38α with NEF reduces PDIA3P1-mediated resistance, along with improved antitumor effects when combined with TMZ [66]. PSMG3-AS1 is upregulated in GBM and linked to the grade of glioma. It promotes TMZ resistance by stabilizing c-Myc in the nucleus. Knockdown increases TMZ sensitivity while overexpression induces resistance [67]. HOTAIRM1 is linked to tumor aggressiveness, radioresistance, and poor prognosis, independent of IDH and MGMT status [100]. Xu Dong Li and colleagues analyzed long non-coding RNA, just proximal to the X-inactive specific transcript (JPX), demonstrating that JPX exerts its GBM-promoting effects through the FTO/PDK1 axis. Their data suggest therapeutic potential for this lncRNA [101]. TUG1, a tumor suppressor-like lncRNA, inhibits cancer stem-cell–like properties and tumorigenicity by downregulating EZH2. In combination with dihydroartemisinin (DHA), its overexpression sensitizes glioma cells towards apoptosis or ferroptosis [102,103]. The H19-miR-93-ATG7 axis plays a role in dopamine agonist treatment of prolactinomas, a possible therapeutic target [81]. HCP5 binds to miR-128 and regulates radiosensitivity of glioma cells and cellular senescence [104]. MVIH silencing in GBM cells results in antitumor effects mediated by the release of miR-302a from the lncRNA sponge [105]. LINC00473, transferred via exosomes, promotes TMZ resistance through a CREB/LINC00473/CEBPα/MGMT axis, suggesting the potential of therapeutic targeting [106]. PVT1 expression is linked to PTEN and EGFR alterations. Knockdown increases TMZ sensitivity via the JAK/STAT pathway, and high expression correlates with shorter survival, indicating prognostic value [107]. The nuclear-enriched lncRNA NEAT1 has been characterized as an oncogenic driver in gliomas, facilitating glioma stem cell proliferation via the NEAT1/let-7g-5p/MAP3K1 signaling axis [108]. It also acts as a ceRNA for miR-324-5p, proposing the NEAT1/miR-324-5p/KCTD20 axis as a therapeutic target [109], and regulates mouse glial cell TMZ resistance via modulation of Connexin 43 [110,111]. In addition, NEAT1 increases β-catenin nuclear transport and H3K27 trimethylation, impacting glioma progression and glycolysis regulation [112]. MALAT1 (NEAT2) is associated with metastasis and drug resistance in gliomas and represents a promising therapeutic target [113,114,115]. lncRNA-ROR induced chemoresistance in nasopharyngeal carcinoma through the p53 pathway suppression [116].

## 3. Discussion

LncRNAs (long non-coding RNAs) are capable of regulating gene expression through their interaction with microRNAs (miRNAs). This regulatory mechanism is of significant interest from a therapeutic standpoint, as the use of lncRNAs allows for the modulation of miRNA activity, which in turn affects cellular signaling pathways involved in numerous biological processes. The ability to intervene in these mechanisms enables direct regulation of crucial pathways for tumor progression, including those that contribute to resistance to therapeutic treatments. In this way, lncRNAs may be a new strategy towards developing innovative therapeutic strategies aimed at restoring or modulating the activity of specific miRNAs, ultimately improving the effectiveness of therapies, combating drug resistance, and enhancing therapeutic response in patients with resistant tumors. Furthermore, these studies have reinforced the role of certain lncRNAs directly associated with the evolution of malignant brain tumors, with effects linked to increased treatment resistance or enhanced sensitivity. Some of the most representative lncRNAs, such as HOTAIR, MALAT1, and MEG3, have been consistently validated over the years, where their role in malignant brain tumors appears evident. However, further in-depth studies that systematically investigate lncRNAs are needed, especially considering their prognostic potential. These studies should explore the use of non-invasive methods, such as liquid biopsy, to assess the prognostic value of lncRNAs. Understanding the functions of lncRNAs in normal tissues is essential for developing their potential as targeted therapies in oncology. As lncRNAs emerge as key players in glioma, their roles in non-malignant contexts must be carefully evaluated. Many lncRNAs demonstrate pleiotropic effects, which are involved in processes like embryonic development, muscle differentiation, metabolism, and immune regulation. For instance, H19 regulates embryonic development, muscle and bone differentiation, adipogenesis, and hair follicle regeneration through ceRNA activity and epigenetic modulation [117]. HOTAIR interacts with chromatin regulators such as PRC2 and LSD1, which play a crucial role in differentiation under physiological conditions [118]. MALAT1, mainly nuclear anchors splicing regulators at nuclear speckles, modulates alternative splicing and transcription in tissues like the endothelium and muscle [119]. MEG3 supports muscle homeostasis, mitochondrial integrity, and osteogenic differentiation via BMP4 and ceRNA-mediated BMP2 regulation [120]. Future studies should not only explore the oncogenic or tumor-suppressive roles of lncRNAs but also consider their pleiotropic effects, especially in terms of tissue-specific expression, compensatory mechanisms, and long-term systemic effects. A thorough understanding of these effects is crucial to ensuring safe and effective lncRNA-based therapies, resulting in minimizing off-target toxicity and accounting for the consequences of their deregulation in tumor progression. Their tissue-specific expression makes lncRNAs potentially useful at a therapeutic level, with the hypothesis that targeting them could lead to fewer side effects. Therefore, understanding the involvement of lncRNAs in drug-resistant mechanisms offers promising avenues for overcoming treatment barriers and improving patient outcomes. After reviewing recent studies on lncRNAs, we have gathered data and confirmation that lncRNAs are principally interesting when they are considered in the context of resistance to therapies such as chemotherapy and radiotherapy. Furthermore, their unique feature lies in the fact that, in cases where their diagnostic, prognostic, and therapeutic roles in relation to resistance are confirmed, it can be stated that, without a doubt, we are dealing with oncogenic lncRNAs. These are generally expressed in tumors but not in healthy tissues. This peculiarity, highlighted in the studies reviewed, is also evident in malignant brain tumors, with a particular focus on tumors such as glioblastoma, one of the main cancers with extremely poor survival prospects. This makes them even more appealing, as it confers a therapeutic potential that is expected to have fewer adverse effects, especially due to their characteristic of exhibiting low or almost no expression in healthy brain tissue. Several studies describe the detection of lncRNA biomarkers in the bloodstream at sites distant from the brain, with relatively high specificity and sensitivity. In some cases, these findings have been validated through imaging data and tissue biopsies. Studies on lncRNAs are interesting, but have noted that studies on experimental therapeutic effects are not present. The role of these lncRNAs at the therapeutic level, in the papers reviewed, seems to only be theoretical at present.

## 4. Conclusion 

In conclusion, we believe it is necessary to encourage and investigate the therapeutic potential of lncRNA inhibitors that contain the positive aspects described above, particularly in tumors, and are linked to the most common pathways associated with therapy resistance, such as PI3K/AKT/PTEN, or as β-catenin in tumor progression. After analyzing the existing literature, we can state that the most reliable data seem to be related to the action of certain lncRNAs in gliomas, such as HOTAIR, NEAT1, MEG3, and MALAT1. Therefore, we hope and are confident to state that the transition from theoretical assertions to data is now necessary.

## Figures and Tables

**Table 1 curroncol-32-00539-t001:** LncRNAs pathway in glioma (prognostic).

Pathway/Mechanism	lncRNA	Key Points / Function	References
MAPK/ERK Pathway	LINC00461 (ECONEXIN)	Promotes glioma progression via MAPK/ERK; affects proliferation, migration, invasion	Yang Y, Ren M [26]
	TPTEP1	Promotes *MAPK14* expression, activates P38 MAPK via antagonizing miR-106a-5p	Tang T, Wang LX [27]
	SNHG12	Sponges miR-129-5p, upregulates *MAPK1 and E2F7*, confers TMZ resistance	Lu C, Wei Y [28]
PI3K/AKT Pathway	LINC00461 (ECONEXIN)	Also involved in PI3K/AKT signaling	Yang Y, Ren M [26]
	MSC-AS1	Knockdown suppresses growth and chemoresistance via PI3K/Akt pathway	Li C, Feng S [29]
	XLOC013218 (XLOC)	Upregulates *PIK3R2* via Sp1, activating PI3K/AKT; promotes TMZ resistance	Zhou J, Xu N [30]
Wnt/β-catenin Pathway	MIR155HG	Knockdown increases TMZ sensitivity by inhibiting Wnt/β-catenin	He X, Sheng J [31]
	SOX2OT	Oncogenic role via Wnt5a/β-catenin signaling	Liu B, Zhou J, Wang C [32]
	HOTAIR	Regulates β-catenin by sponging miR-214; influences TMZ sensitivity	Lan T, Quan W [33]
miRNA Sponge/ceRNA	SNHG12	Sponges miR-129-5p, upregulates *MAPK1/E2F7*	Lu C, Wei Y [28]
	ECONEXIN (LINC00461)	Sponges miR-411-5p regulating TOP2A	Deguchi S, Katsushima K [34]
	NCK1-AS1	Modulates miR-137/TRIM24, increasing TMZ resistance	Chen M, Cheng Y [35]
	LINC01123	Sponges miR-151a, upregulates *CENPB*; increases radioresistance	Tian W, Zhang Y [36]
	HOTAIR	Sponges multiple miRNAs (miR-214, miR-526b-3p, miR-519a-3p, miR-125) involved in drug resistance and progression	Lan T et al. [33], Wang X et al. [37], Yuan Z et al. [38], Zhang J et al. [39]
Drug Resistance/ Chemoresistance	MSC-AS1	Knockdown reduces TMZ resistance via miR-373-3p/CPEB4 and PI3K/Akt	Li C, Feng S, Chen L [29]
	SNHG12, NCK1-AS1, LINC00520	Associated with TMZ resistance	Lu C, Wei Y [28]
	XLOC013218 (XLOC)	Activates TMZ resistance through PI3K/AKT signaling	Zhou J et al. [40]
	MAFG-AS1	Decreases radiosensitivity via miR-642a-5p/Notch1 axis	Zhang X, Li R [41]
	FOXD3-AS1	Critical for GBM survival and TMZ resistance	Ling Z, Zhang J [42]
	ZBED3-AS1	Linked to TMZ resistance and glycolysis	Dong J, Peng Y [43]
	ATXN8OS	Mediates ferroptosis and TMZ resistance via ADAR/GLS2 pathway	Luo J, Bai R [44]
Cell Cycle / Proliferation	LINC00021	Represses p21, promoting glioblastomagenesis	Zhang S, Guo S [45]
	E2F1	Targets promoter of *LINC00021*	Zhang S, Guo S [45]
	HOTAIR	Regulates cell cycle, promotes invasion and metastasis	Zhang et al. [45], Botti G et al. [46]
EMT	MAFG-AS1	Promotes EMT, proliferation, metastasis	Zhang X, Li R [41]
	MEG3	Inhibits EMT in some cancers; promotes EMT in others	Degirmenci Z [47]
	HOTAIR	Promotes EMT and invasion	Zhang J, Chen G [39]
Other Mechanisms	HOXA-AS2	Knockdown reduces IGF1 expression via miR-302a-3p	Lin L, Lin D [48]
	HOTAIR	Delivered by extracellular vesicles/exosomes, regulates miRNA axes, promotes TMZ resistance	Wang X et al. [37], Yuan Z et al. [38]
	MEG3	Stimulates p53 transcriptional activity; regulates proliferation, apoptosis, EMT	Wang X, Li Z [49]

**Table 2 curroncol-32-00539-t002:** LncRNAs pathway in glioma (therapeutic).

Pathway/Mechanism	Main lncRNAs Involved	Main Effects	Key References
NF-κB	LIFR-AS1, LINC01057, H19	Apoptosis, MES differentiation, resistance	Ding et al. [60], Duan S et al. [61]
miRNA-ceRNA	KCNQ1OT1, NCK1-AS1, TP53TG1, PSMB8-AS1, CCAT2, LINC00174, etc.	Proliferation, chemoresistance, radioresistance	Wang W et al. [62], Hu T et al. [63], Gao W et al. [64]
Chemoresistance (TMZ)	KCNQ1OT1, BC200, PDIA3P1, PSMG3-AS1, LINC01410, etc.	Resistance, proliferation	Wang W et al. [62], Su YK et al. [65], Gao Z et al. [66], Zhou L et al. [67]
PI3K/Akt/mTOR	CRNDE, MAGI2-AS3, LINC01410	Autophagy, TMZ resistance	Zhao Z et al. [68], Fu T et al. [69]
Autophagy	Linc-RA1, CRNDE, H19	Radio and chemo resistance	Zhao Z et al. [68], Zheng J et al., [70]
Wnt/β-Catenin	RMRP, LINC00511	TMZ resistance, proliferation	Liu T et al. [71], Lu Y et al. [72]
Radioresistance	Linc-RA1, NCK1-AS1, TP53TG1, RBM5-AS1, HCP5, LINC01123	Radiotherapy resistance	Zheng J et al. [70], Zhu C et al. [73], Wang B et al. [74], Gao W et al., [64]
DNA Damage/Repair	EPIC1, EPIC-0628, CHK1, DARS1-AS1	Cell cycle, DNA repair, TMZ sensitivity	Hu T et al. [63], Liao X et al. [75], Yang E et al. [76], Zheng C et al. [77]
EMT/Invasion	lncRNA-MUF, H19, XIST	EMT, tumor invasion	Shree B et al. [78], Zhao J [30]

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
