# Peer review of "The Role of LncRNAs in Radio- and Chemoresistance of Glioblastoma: Prognostic or Therapeutic?"

_curroncol, 2025, doi:10.3390/curroncol32100539_

Round 1
Reviewer 1 Report
Comments and Suggestions for Authors
Dear Authors,
I have read your manuscript by the title "The role of lncRNAs in radio- and chemoresistance of gliomas: prognostic or therapeutic? " and I am hereby sharing with you my synthetic considerations.
I would like to commend you for the topic pick and for the effort invested in organizing and presenting data about a wide spectrum of lncRNAs.
However, since your goal is to advocate for more translational research, I feel it would be beneficial to the reader to include also a few thoughts about pleiotropic effects of lncRNAs, and known/plausible-yet-unknown side effects to watch out for when designing future studies. What is the role of the studied lncRNAs in normal tissues? Are there data from domains outside of oncology that the future researchers should be aware of?
I look forward to reading an update version of your manuscript.
Keep up the good work. Best regards,
Comments on the Quality of English Language
The manuscript might benefit from a review by a copywriter, as several sentences (one example for all, the closing sentence "...are confident that the transition from theoretical assertions to data is now necessary") are formulated in such a way that might pass as a mechanical translation from a different language, deforming/modifying the message for the average reader.
Reviewer 2 Report
Comments and Suggestions for Authors
To authors
First of all, I would like to give the authors congratulation on the successful study showing the role of lncRNAs in tumor progression and therapy resistance in malignant brain tumor, such as glioblastoma.
The authors' review paper focuses on RNA interference among epigenetic mechanisms, and its role in malignant brain tumors was well identified. As it deals with relatively limited and small topics, it seems that the contents have been handled well in depth. It's a well-written review paper that doesn't have much to point out, but if some expressions are misleading, it will be more helpful for readers to understand.
For example, since the research area of epigenetics is not much done in actual benign brain tumors and targets various biological mechanisms of malignant brain tumors, it feels more concentrated to write "CNS cancer" or "malignant brain tumor" rather than the broad term of "brain tumor" as in line 21. I think authors need to rewrite these obscure words throughout the paper.
Also, I think it may be more natural to briefly explain the main mechanism of cancer epigenetics and to mention the lncRNA story among them, rather than suddenly showing the lncRNA in the introduction. I would like to give my opinion on how to briefly describe three main mechanisms of epigenetic regulation in the introduction.
In terms of terms, the expression "prognostic role of lncRNA" is easier to understand than the expression "prognostic lncRNA" written in the subtitle of 2.1.1. The expressions used by the authors seem to be somewhat exaggerated and misleading, which may make the readers uncomfortable. The same goes for the expression "therapeutic lncRNA." I think the expression "lncRNA as a therapeutic target" is more appropriate.
Any other sentence or statement is considered to be within the acceptable range.
Good Luck.
